RNA-Seq and genetic diversity analysis of faba bean (Vicia faba L.) varieties in China

Hou Wanwei 1 2
Zhang Xiaojuan 3
Liu Yuling 2
Liu Yujiao 2
Feng Bai li fengbaili@nwsuaf.edu.cn 1
1 College of Agronomy, Northwest A&F University , Yangling, Shaanxi , China
2 Qinghai Academy of Agricultural and Forestry Sciences, Qinghai University , Xining, Qinghai , China
3 College of Eco-Environmental Engineering, Qinghai Universit , Xining, Qinghai , China
Azeem Farrukh
Electronic publication date: 2023 Jan 10
Publication date: 2023
Volume: 11
Electronic Location ID: e14259
Received 2022 Apr 29; Accepted 2022 Sep 27
Copyright: ©2023 Hou et al.
Copyright year: 2023
Copyright holder: Hou et al.
License: This is an open access article distributed under the terms of the Creative Commons Attribution License, which permits unrestricted use, distribution, reproduction and adaptation in any medium and for any purpose provided that it is properly attributed. For attribution, the original author(s), title, publication source (PeerJ) and either DOI or URL of the article must be cited.
License URL: https://creativecommons.org/licenses/by/4.0/

Keywords: Vicia faba L, RNA-Seq, Gene function analysis, Simple sequence repeats (SSR) markers

Funding: Qinghai Applied Fundamental Research Program 2021-ZJ-744 China Agriculture Research System CARS-08 This research was funded by Qinghai Applied Fundamental Research Program, grant number 2021-ZJ-744 and the China Agriculture Research System, grant number CARS-08. The funders had no role in study design, data collection and analysis, decision to publish, or preparation of the manuscript.

==============================
Background

Faba bean (Vicia faba L) is one of the most important legumes in the world. However, there is relatively little genomic information available for this species owing to its large genome. The lack of data impedes the discovery of molecular markers and subsequent genetic research in faba bean. The objective of this study was to analyze the faba bean transcriptome, and to develop simple sequence repeat (SSR) markers to determine the genetic diversity of 226 faba bean varieties derived from different regions in China.

Methods

Faba bean varieties with different phenotype were used in transcriptome analysis. The functions of the unigenes were analyzed using various database. SSR markers were developed and the polymorphic markers were selected to conduct genetic diversity analysis.

Results

A total of 92.43 Gb of sequencing data was obtained in this study, and 133,487 unigene sequences with a total length of 178,152,541 bp were assembled. A total of 5,200 SSR markers were developed on the basis of RNA-Seq analysis. Then, 200 SSR markers were used to evaluate polymorphisms. In total, 103 (51.5%) SSR markers showed significant and repeatable bands between different faba bean varieties. Clustering analysis revealed that 226 faba bean materials were divided into five groups. Genetic diversity analysis revealed that the relationship between different faba beans in China was related, especially in the same region. These results provided a valuable data resource for annotating genes to different categories and developing SSR markers.

Introduction

Faba bean (Vicia faba L.) is one of the most important crops worldwide. It is grown in China, Ethiopia, Egypt, the Andean States of South America, Europe, and Australia (Duc et al., 2010). Approximately 4.8 million tons of faba beans were grown in 2017, which ranks fourth in production among cool-season crops (FAOSTAT, 2018). Faba beans are a good source of protein for humans, especially those in West Asia and North Africa, and is an affordable livestock feed in developed regions, such as Europe and Australia (Alghamdi et al., 2012; Johnston et al., 1999a; Johnston et al., 1999b). Understanding the soil nitrogen fixation of faba bean crop rotation is important for improving green agriculture practices for environmental preservation (Stoddard et al., 2009).

Faba bean is a partially allogamous diploid (2n = 12) species (Raina & Ogihara, 1995), with a genome size of approximately 13 Gb, which is the largest genome among legume crops (Bennett & Smith, 1982; Johnston et al., 1999a; Johnston et al., 1999b). Although this is an important crop, there is less attention focused on faba bean than other legumes owing to its genome size and six pairs of remarkably large chromosomes.

Research on the faba bean genome is challenging in comparison with other legumes. However, considerable progress has been made in the mapping of genes using genetic diversity analysis for crops with important agronomy characteristics. Previous genetic linkage maps were constructed with different populations and molecular markers. The initial molecular markers of faba bean focused on fragment length polymorphisms (RFLPs) (Van de Ven et al., 1991; Torres, Weeden & Martín, 1993), and randomly amplified polymorphic DNA (RAPDs) (Link et al., 1995; Avila et al., 2003). The first linkage map of faba beans was constructed with RFLP, RAPD and isozyme markers by Van de Ven et al. (1991). Second generation molecular markers, such as simple sequence repeats (SSRs) (Pozarkova et al., 2002; Roman et al., 2004), amplified fragment length polymorphisms (AFLP) (Zeid, Schon & Lin, 2003; Zong et al., 2009; Zong et al., 2010), sequence-characterized amplified regions (SCAR) Gutierrez et al. (2007), and inter-simple sequence repeats (ISSR) (Wang et al., 2012; Hou et al., 2018) made high density linkage maps and genetic diversity analysis possible. For example, an integrated approach was first used to develop SSR markers from selected regions of the faba bean (Pozarkova et al., 2002), which were then mapped by Roman et al. (2004). Different spring and winter germplasms derived from China and abroad were compared with AFLP markers (Zong et al., 2009; Zong et al., 2010).

With the rapid development of next-generation sequencing technology (NGS), numerous expressed sequence tags (ESTs) have become publicly available and provide useful information for EST-based markers (Ma et al., 2011; Kaur et al., 2012a; Kaur et al., 2012b; Yang et al. , 2012; Beier et al., 2017; Mokhtar & Atia, 2019). This has provided a sound foundation for comparative genome analysis with other legume species (Chapman et al., 2009). The second-generation sequence technology has been foundation for sequencing faba bean genomes and researchers have made some initial sequencing efforts. Additional high density physical map and map-based cloning is possible for faba bean. Ma et al. (2011) developed 21 EST-derived SSRs markers; these polymorphic markers were used for gene tagging and MAS in faba bean. Yang et al. (2012) released high-throughput microsatellite markers using next generation sequencing. Kaur et al. (2014) first reported high-density genetic mapping with SNP-based markers in faba bean ascochyta blight resistance. The first consensus map for the faba bean contained 687 SNP markers on six linkage groups, each presumed to correspond to one of the faba bean chromosomes (Webb et al., 2016). Density enhancement genetic linkage map of the faba bean was conducted by screening 5,325 SSR genomic primers and 2,033 EST-SSR primers (Yang et al., 2019). These are useful for marker-assisted selection and breeding in this important legume crop. Carrillo-Perdomo et al. (2020) reported the most saturated consensus genetic map with 1,728 SNP markers distributed in six linkage groups using three mapping populations. Recently, Mokhtar et al. (2020) released the Vicia faba omics database (VfODB), which included ESTs, EST-SSRs, mtSSRs, microRNA-target markers and genetic maps in the faba bean. This powerful database will be a helpful platform for the molecular breeding of the faba bean.

However, there are not many identified faba bean markers that can be used in genetic studies and there is an urgent demand to develop various kinds of molecular markers to enhance marker development and breeding strategies in the future. To date, RNA-seq technology has been an effective tool to generate genome-wide transcriptome profiles and obtain substantial SSRs, especially for species whose genome has not been sequenced (Garg & Jain, 2013; Zhang et al., 2011; Xiao et al., 2013).

RNA-seq has been successfully used in various species, including rice (Xu, Gao & Wang, 2012), maize (Davidson et al., 2011), chickpea (Garg et al., 2011), field pea (Sudheesh et al., 2015), and faba bean (Yang et al., 2020). RNA-Seq will provide a feasible approach to develop SSR markers for both genetic diversity and mapping analysis of the huge, non-sequenced faba bean genome.

Previous studies have reported EST-SSR markers in faba bean, however, the numbers that can be used in genetic analysis and linkage mapping are still limited. In order to conduct genetic analysis, we first used four faba bean varieties for RNA sequencing; this data was used (1) to obtain faba bean transcriptome information and get a better understanding of the annotated genes, (2) to develop SSR markers and validate their polymorphisms, (3) to verify the relationship between different faba bean germplasms in China.

Materials and Methods

Plant materials

Four different varieties of faba bean, Qinghai11 (indeterminate growth), Qingcan16 (determinate growth), Qingcan 18 (zero tannin content) and YL01 (high tannin content) with extreme phenotype, were selected for transcriptome sequencing. The four varieties were grown in the experimental station of the Qinghai Academy of Agricultural and Forestry Sciences. The young leaves of different varieties were collected for three biological repeats and the RNA was extracted.

A total of 226 samples (Table S1) derived from different regions of China were used for the development of SSR markers and genetic diversity research. The 226 accessions were the main varieties or special germplasm resources derived from different provinces of China. These accessions will be used for the construction of the core germplasm for the China faba bean in the future. Among these, 63 varieties were collected from the northwest, 79 varieties were sampled from the southwest, 15 varieties were found in northern China, 54 varieties came from eastern China, three varieties were found in southern China, and the remaining 12 varieties were distributed in central China.

The 226 samples were provided by the Qinghai Academy of Agriculture and Forestry Sciences and were planted in the experiment field in Qinghai in March 2020.

RNA extraction and cDNA library construction

RNA extraction and cDNA library construction samples were isolated from the young leaves of four different varieties: Qinghai11, Qingcan16, Qingcan 18, and YL01. The total RNA was extracted using Trizol reagent (Invitrogen, Carlsbad, CA, USA) according to the manufacturer’s instructions and were stored at −20 °C for later analysis. DNase I was used to treat the RNA samples to remove DNA contamination. RNA samples were mixed according to the equivalent concentration of each variety. The concentration and integrity of the total RNA was examined and degraded using a 2100 BioAnalyzer (Agilent, Santa Clara, CA, USA) and agarose gel electrophoresis. The total RNA was processed using oligo (dT) magnetic beads. The selected RNA was used to create a full-length cDNA library. An Agilent 2100 BioAnalyzer and ABI Steponeplus real-time PCR system were used to test the constructed libraries. The resulting products were used for sequencing.

Data filtering and de novo assembly

The Illumina NovaSeq 6000 platform was used for paired-end sequencing. Raw reads were filtered via SOAPnuke (v1.4.0) (Chen et al., 2017a; Chen et al., 2017b). The reads were first preprocessed to eliminate joint contamination. The low-quality reads were discared (reads with ambiguous N bases >5% and more than 15% of bases Q <20 bases). Clean data were obtained after removing adapter-containing, poly-N-containing, and low-quality reads from the raw data. The Q20, Q30, and the GC content of the clean data were calculated.

All downstream analyses were based on high-quality clean data. De novo transcriptome assembly was performed using Trinity (v2.0.6) (Grabherr et al., 2011) (https://github.com/trinityrnaseq/trinityrnaseq/wiki) with the inchworm k-mer method. Tgicl (v2.0.6) (Pertea et al., 2003) was used to perform clustering and eliminate redundant data in the assembled transcripts to obtain unique genes. The sequences without redundancy, containing the fewest Ns, and not being extended on either end, were defined as unigenes. The assembled transcripts were processed for further expression analysis and functional annotation. Clean reads were mapped to the assembled unique genes by Bowtie2 (v2.2.5) (Langmead & Salzberg, 2012) (https://bowtie-bio.sourceforge.net/bowtie2/index.shtml) and the expression level of genes were calculated by RSEM (v1.2.8) (Bo & Dewey, 2011) and normalized to fragments per kilo base of transcript per million mapped reads (FPKM).

The raw data was also calculated by RNASeqPower (https://bioconductor.org/packages/release/bioc/html/RNASeqPower.html) (Hart et al., 2013). The sequencing depth of RNA Seq-Power package in R was 32 and the sample number was 3 (Table S2).

The sequencing data were then submitted to the National Center for Biotechnology Information (NCBI) database with the following accession numbers: PRJNA823658; SAMN27335281; SAMN27335282; SAMN27335283; SAMN27335284; SAMN27335285; SAMN27335286; SAMN27335287; SAMN27335288; SAMN27335289; SAMN27335290; SAMN27335291; SAMN27335292.

Unigene function annotation

Unigene function information was performed with bioinformatics methods based on data from various databases. The databases were as follows: (1) NR, diamond 0.8.22 with an E-value ≤ 1.0E–5, (https://github.com/bbuchfink/diamond) (Benjamin, Chao & Daniel, 2014); (2) KOG, diamond 0.8.22 with an E-value ≤ 1.0E–3 (Benjamin, Chao & Daniel, 2014); (3) SwissProt, diamond 0.8.22 with an e-value = 1e–5) (Benjamin, Chao & Daniel, 2014); (4) NT, NCBIBLAST2.2.28+ (ftp://ftp.ncbi.nlm.nih.gov/blast/executables/LATEST/) (Altschul et al., 1997); (5) PFAM, HMMER 3.0 package and hmmscan with an e-value = 0.01, (https://www.ebi.ac.uk/Tools/hmmer/) (Alex et al., 2004); (6) KO, KEGG automatic annotation server with an e-value=1e–10 (https://www.genome.jp/tools/kaas/) (Moriya et al., 2007) and (7) GO, Blast2GO 2.5 and a custom script with an e = value = 1e–6 (https://www.blast2go.com) (Götz et al., 2008).

SSR marker development and validation

A microsatellite identification tool (Microsatellite identification tool, MISA, http://pgrc.ipkgatersleben.de/misa/) was used to detect the SSR loci from all sequences. The screening criteria for SSR loci were the minimum repeats of mono-10, di-6, tri-5, tetra-5, penta-5, and hexa-5. Selected microsatellite loci were used to design SSR markers using Primer 3.0 (https://bioinfo.ut.ee/primer3/) based on the flanked sequence (Rozen & Skaletsky, 2000). The basic parameters were as follows: primer length 15–22 bp, Tm of 40–60 °C and the PCR product length in the range of 100–300 bp.

The genomic DNA from 226 faba bean varieties was extracted according to the DS (sodium lauroylsarcosine) protocol (Song & Langridge, 1991; Song & Henry, 1994). DNA quality and quantity were tested by 1.5% agarose gel electrophoresis. The developed SSR markers were first verified using 50 randomly-selected faba bean accessions, including 16 varieties from the Qinghai province, 20 varieties from the Yunnan province, 10 varieties from the Sichuan province, and four varieties from the Jiangsu province.

PCR was carried out in a S1000 Thermal Cycler (BIO-RAD, Hercules, CA, USA) in volumes of 10 µL. The PCR system contained 4.0 µL 2 ×PCR Master Mix (TaKaRa, Shiga, Japan), 2.0  µL DNA (10 ng/ µL), 1.0 µL of each primer pair (2.0 pmol/µL), and 2.0 µL ddH2O. The PCR conditions were as follows: denaturation at 94 °C for 5 min, followed by 30 cycles of 94 °C for 30 S, 55 °C for 30 s, 72 °C for 30 S, and a final extension for 10 min at 72 °C. PCR products were heated and denatured at 95 °C for 5 min. A total of 5 µL of each sample was loaded with a 6% denaturing polyacrylamide gel using the silver staining method (Bassam, Anolles & Gresshoff, 1991).

Genetic diversity analysis

SSR data were scored according to the migration rate of the PCR amplified fragments at each SSR locus and the data format was converted according to the requirements of the analysis software. Popgene V1.32 software was used to calculate the allele number (NA) and effective number of alleles (NE). The polymorphism information content (PIC) and Nei’s genetic diversity (H) were calculated using Power Marker V3.25 software. The genetic diversity analysis of faba bean was tested using Nei’s genetic distance and the unweighted pair group method using averages (UPGMA) (non-weighted averages) method.

The genetic structure of the 226 faba beans were analyzed with Structure 2.3.4 (Pritchard, Stephens & Donnelly, 2000; Falush, Stephens & Pritchard, 2003). The parameters used were as follows: length of burn-in period = 10,000, number of MCMC reps after burn-in = 10,000, number of populations (K) = 1–10, number of iterations = 10. The optimal population structure and subgroup number were determined using the delta K (ΔK) value (Evanno, Regnaut & Goudet, 2005) according to the Structure Harvest website (http://taylor0.biology.ucla.edu/struct_harvest/) (Earl & VonHoldt, 2012).

Results

RNA sequencing and sequence annotation

A total of 92.43 Gb of paired-end raw data were obtained from the 12 faba bean samples (four accessions with three replicates) using the Illumina NovaSeq 6000 platform. A total of 89.31 Gb of clean reads were obtained after filtering the reads for quality. The Q20 percentage of the reads was over 98% and the Q30 percentage was over 94% (Table S3). All clean reads were assembled with Trinity and then Tgicl was used to cluster the transcripts to eliminate redundancy and obtained the initial unigene. Tgicl was used for multiple samples to cluster the unigene of each sample to eliminate redundancy and obtain the final unigene for subsequent analysis (Table S4). After clustering, the quality indexes and the length distribution of the unigene are shown in Table S4. A total of 133,487 unigenes were finally obtained with a GC content of 38.77%, and the mean length of the assembled unigenes was 1,334 bp (N50 = 1,858 bp).

The 133,487 unigenes were composed of 30,443 unigenes (22.8%) with lengths of 200 to 500 bp; 30,887 unigenes (23.1%) with lengths of 500 to 1000 bp; 26,018 unigenes (19.5%) with lengths of 1,000 to 1,500 bp; 19,082 unigenes (14.3%) with lengths of 1,500 to 2,000 bp; and 27,057 unigenes (20.3%) with lengths longer than 2,000 bp (Fig. 1A).

TransDecoder software was used to identify candidate coding regions (CDS) in the unigenes. The longest open reading frame was extracted, and then the Pfam protein homologous sequences were searched by Blast comparison between the SwissProt database and Hmmscan to predict the CDS. The predicted total number of CDS was 71,282 with a GC content of 41.86%, and CDS length from 12,783 bp to 297 bp (N50 = 1,359 bp).

A total of 71,282 CDS were composed of 19,091 CDS (26.8%) with lengths of 200 to 500 bp; 24,053 CDS (33.7%) with lengths of 500 to 1,000 bp; 13,916 CDS (19.5%) with lengths of 1,000 to 1,500 bp; 6,961 CDS (9.8%) with lengths of 1,500 to 2,000 bp; and 7,261 CDS (10.2%) with lengths longer than 2,000 bp (Fig. 1B).

Gene function classification

Gene function annotation resulted in 133,487 unigenes. The numbers and percentage of annotation results for the seven databases, including NR, NT, PFAM, Swissprot, GO, KEEG and KOG, are shown in Table 1 (Fig. 2A). The largest number of genes (88,629; 66.40%) were annotated by NR, and the fewest were annotated by Pfam, (62,986; 47.19%) (Fig. 2A). Venn diagram of gene annotation with five databases, including NR, NT, PFAM, GO, and KOG, showed that a total of 44,188 genes could be annotated (Fig. 2B).

Figure 1 The number and distribution frequency of unigenes and CDS.

(A) Unigenes; (B) CDS. The x-axis shows the sequence sizes and the y-axis indicates the numbers.

After GO annotation, the unigenes were classified into three categories: biological process, cellular component, and molecular function (Fig. 3A). Among these, 83,391 (46.9%) unigenes for molecular function accounted for the majority of the unigenes, followed by 52,210 (29.3%) unigenes for cellular components, and 42,359 (23.8%) unigenes for participating in biological processes, covering a comprehensive range of GO categories. In the molecular function category, catalytic activity (39,008) and binding (35,451) were the two main categories. Moreover, 4,478 unigenes were assigned to transporter activities. However, few unigenes were involved in transcription regulator (1,482), structural molecule activity (1,404), molecular function regulator (859), and antioxidant activity (419). The rest of the unigenes were related with cargo receptor activity, molecular carrier activity, nutrient reservoir activity, and protein tag and translation regulator activity. Among the cellular components, unigenes of membrane part (21,296), cell (18,105), and organelle part (9,888) were the dominant unigenes. However, only a few unigenes were involved in the membrane (860), extracellular region (776), protein-containing complex (523), virion part (393), and symplast (283). The remaining unigenes participated in the supramolecular complex, organelle, and nucleoid. The biological processes unigenes were as follows: unigenes of the cellular (18,246), biological regulation (7,015), cellular component organization (4,664) processes, metabolic (4,025) and localization (3,985) accounted for the majority of the unigenes. However, only a few unigenes were determined to be a to response to stimulus (2,196), multicellular organismal process (1,091), and multi-organism process (717). The remaining unigenes were associated with the other biological process, including rhythmic process, nitrogen utilization, and growth.

Table 1 Numbers and percentage of unigenes with different databases.

Different numbers and percentage.

Values	Number	Percentage (%)	
Total	133,487	100	
NR	88,629	66.4	
NT	79,861	59.83	
Swissprot	64,270	48.15	
KEGG	69,648	52.18	
KOG	68,068	50.99	
Pfam	62,986	47.19	
GO	72,557	54.36	

Figure 2 Genes annotation with NT, PFAM, KOG, and NR database and Venn diagram of gene annotation with seven (NR, NT, Swissprot, KEGG, KOG, Pfam and GO) databases.

(A) Genes annotation with four databases; The x- axis shows the seven databases and the y- axis indicates the numbers of gene annotation. (B) venn diagram with seven databases. The different color represented different databases.

Figure 3 Gene classification with GO, KOG and KEEG.

(A) GO annotation of the unigenes; the results are summarized in three main categories: biological process, cellular component, and molecular function; (B) KOG functional classification of the faba bean unigenes; (C) histogram of the KEGG classification of the assembled unigenes in faba bean. Each color showed different function.

A total of 68,068 KOG-annotated unigenes were classified according to KOG group and divided into 25 groups after KOG annotation (Fig. 3B). Among these genes, the number associated with only the general function prediction was the highest (15,721); the number in the signal transduction mechanisms ranked second (7,707); and the number associated with function unknown ranked third (5,771).

BlastX was used to match a single gene to KEGG to assess the inclusiveness of the transcriptome library and the efficiency of the annotation program. The results showed that a total of 69,648 unigenes were classified according to their participation in KEGG metabolic pathways, including cellular processes (3,374), environmental information processing (4,361), genetic information processing (15,477), metabolism (42,034), and organismal systems (2,405) (Fig. 3C). The largest number of genes were the global and overview maps, which involved in the metabolism branch, with 15,875 genes.

According to the E-value distribution in the NR database, 88,574 genes were found to be related to genes annotated in other species (Fig. 4), of which 34,014 (38.4%) genes were related to Medicago truncatula, and 15,617 (17.6%) genes were similar to Cicer Arietinum. A total of 14,010 (15.8%) genes were related to Trifolium pratense, 12,516 (14.1%) genes were related to Trifolium subterraneum, and 2,009 (2.2%) genes were related to Pisum sativum. A total of 2,623 transcription factor-related genes were detected by searching the NR datasets (Fig. 5), which were divided into 58 categories. The similarity genes included 318 MYB-related genes, 219 BHLH-related genes, 172 C3H-related genes, 169 AP2-EREBP-related genes, 128 WRKY-related genes, and 117 NAC-related genes.

Figure 4 Similarity of the unigenes in the NR dataset.

Pie chart of unigene similarity search against the NR dataset. E-value distribution of the BLAST hits for each unigene, with an E-value threshold of 10–5 in the NR database.

Figure 5 Number of unigene similarity search against the NR dataset.

Histogram of the unigenes in faba bean with NR databases analysis . The x- axis shows the number of genes and the y- axis indicates the 58 categories.

SSRs frequency and distribution

Among the 133,487 unigenes obtained in this study, 16,202 (12.1%) unigenes contained one or more SSR repeats, 1,693 (1.3%) unigenes contained at least two independent SSR repeats, and 268 (0.2%) contained compound SSRs of different motifs (Table 2). These compound SSRs were distributed randomly in various genic SSR units. Among them, the SSRs included mono-nucleotides (9,370), di-nucleotides (3,087) tri-nucleotides (3,458), tetra-nucleotides (163), hexa-nucleotides (83), penta-nucleotides (41), and others (24). The mono-nucleotides were the most abundant (9,370; 60.1%), followed by the tri-nucleotides (3,458; 21.3%). SSR repeats per locus ranged from five to 83. The most common repeats occurred more than 10 times, and the subsequent repeats occurred 10 times, five times and six times. The (A/T) n repeats were the most abundant nucleotide repeats (98.8%) (mono-nucleotide motif). The other six main motif types was the (AG/CT) n di-nucleotide repeat (74.3%), (AAC/GTT) n tri-nucleotide repeat (23.7%), (AAAT/ATTT) n tetra-nucleotide repeat (38.6%), (AGATG/ATCTC) n penta-nucleotide repeat (14.6%), and the (AATCAT/ATGATT) n hexa-nucleotide repeat (4.8%).

Table 2 SSRs repeat motif and numbers.

Summarizing of different repeats and numbers.

Repeat motif	No. of repeats	
	5	6	7	8	9	10	>10	Total	
Mono-nucleotide (9,370)									
A/T	–	–	–	–	–	3,948	5,313	9,261	
C/G	–	–	–	–	–	22	87	109	
Di-nucleotide (3,087)									
AC/GT	–	170	89	44	21	13	27	364	
AG/CT	–	636	471	345	238	143	462	2,295	
AT/AT	–	168	70	67	31	22	66	424	
CG/CG	–	2	2					4	
Tri-nucleotide (3,458)									
AAC/GTT	478	169	105	50	5	3	8	818	
AAG/CTT	461	163	58	23	3	5	12	725	
AAT/ATT	237	92	77	50	7	7	65	535	
ACC/GGT	184	90	44	22	7	4	3	354	
ACG/CGT	13	2	1					16	
ACT/AGT	51	21	6	4			1	83	
AGC/CTG	61	21	3	1			1	87	
AGG/CCT	81	39	18	3	1	1	2	145	
ATC/ATG	317	147	80	51	14	11	35	655	
CCG/CGG	28	5	5	1	1			40	
Tetra-nucleotide (163)									
AAAC/GTTT	7	3	2	1				13	
AAAG/CTTT	9	10	1	2				22	
AAAT/ATTT	44	9	3	1	1	2	3	63	
AATT/AATT	18	4	1	1				24	
ACAT/ATGT	8							8	
other	20	7	1	1		1	3	33	
Penta-nucleotide (41)									
AAAAC/GTTTT	4							4	
AAAAT/ATTTT	3							3	
AAGAG/CTCTT	4							4	
AGATG/ATCTC	4	1					1	6	
AAATT/AATTT	2							2	
other	14	1	3	2		2		22	
Hexa-nucleotide (83)									
AAAAAC/GTTTTT	3							3	
AACCCT/AGGGTT	4							4	
AACCTC/AGGTTG	3							3	
AACGAC/CGTTGT	2		1					3	
AATCAT/ATGATT	3	1						4	
other	24	21	6	5	3	3	4	66	
total	2,087	1,782	1,047	674	332	4,187	6,093	16,202	
%	12.9	11.0	6.5	4.2	2.0	25.8	37.6		

SSR polymorphism analysis

A total of 5,200 EST-SSR markers were developed from the 133,487 unigenes after filtering (Table S5). To validate their polymorphisms for faba bean accessions, one set of 200 markers was chosen from the above list of loci. Among the selected 200 EST-SSR markers, 103 pairs of SSR markers produced clear amplicons with the expected sizes in nearly all of the 226 faba bean varieties. Among the successful markers, 103 (51.5%) polymorphic genic SSR markers were identified, containing 39 di-, 47 tri-, two tetra-, and one penta-, while the other 14 markers were monomorphic.

A total of 103 polymorphic EST-SSR markers were then used to test the polymorphism of 226 faba bean varieties (Fig. 6; Table S6). A total of 498 polymorphic loci were scored with an average number of 4.84 alleles per locus. The most abundant polymorphism locus was T_DN30600 with 10 alleles. Only two alleles were detected for T_DN20824, T_DN32959, T_DN27103, T_DN31063, T_DN24924, T_DN15279, T_DN34099, and T_DN19478.

Figure 6 PAGE gel of SSR marker T_DN20824 among 64 faba bean varieties.

M: pBR322 DNA/MspI DNA ladder; 1-64: different faba bean varieties (the 64 varieties are listed in Table S1).

The polymorphism information content (PIC) varied from 0.1273 (T_DN34099) to 0.7913 (T_DN28326), with an average value of 0.4959 (Table 3). The proportion of the PIC value was greater than the average proportion of 47.57%. We observed that the effective number of alleles (Ne), Shannon’s index (I), observed heterozygosity (Ho) and expected heterozygosity (He) were 2.4581, 1.0278, 0.0122 and 0.5512, respectively. The number of effective alleles (Ne) was distributed from 1.158 to 5.372 and the average number was 2.4581. The Shannon’s index (I) was arranged from 0.263 to 1.902 with an average of 1.0278. The mean values of observed heterozygosity (Ho) and expected heterozygosity (He) were 0.0122 and 0.5512, respectively. The results indicated that the selected markers showed high diversity between faba bean varieties and could be used for further genetic analysis.

Table 3 Analysis of 103 polymorphism SSR markers.

Polymorphism analysis of developed SSR markers.

Marker	Na	Ne	I	Ho	He	PIC value	
T_DN32469	5	2.6330	1.1500	0.0000	0.6200	0.5494	
T_DN24459	8	3.0370	1.4750	0.0000	0.6710	0.6421	
T_DN27482	5	1.7990	0.8600	0.0000	0.4440	0.4035	
T_DN22376	6	2.2300	1.0030	0.0000	0.5520	0.4802	
T_DN29279	8	3.2450	1.3840	0.0000	0.6920	0.6465	
T_DN1835	4	1.9720	0.7950	0.0000	0.4930	0.4000	
T_DN33317	3	1.9770	0.8280	0.0000	0.4940	0.4262	
T_DN24385	5	2.2900	1.1570	0.0000	0.5630	0.5345	
T_DN26689	6	2.9240	1.2840	0.0000	0.6580	0.6028	
T_DN23895	6	2.7080	1.1960	0.0000	0.6310	0.5661	
T_DN31712	5	3.2060	1.3680	0.0000	0.6880	0.6482	
T_DN20824	2	1.8800	0.6610	0.0000	0.4680	0.3586	
T_DN28915	7	1.5950	0.7980	0.0000	0.3730	0.3481	
T_DN32959	3	1.7050	0.6250	0.0000	0.4140	0.3318	
T_DN24924	5	1.7430	0.6840	0.0000	0.4260	0.3474	
T_DN27054	3	2.7520	1.0500	0.0000	0.6370	0.5599	
T_DN27054	5	2.2520	1.1170	0.0000	0.5560	0.5206	
T_DN32461	3	1.9130	0.6930	0.0000	0.4770	0.3681	
T_DN32430	7	2.7750	1.2950	0.0000	0.6400	0.5948	
T_DN32302	6	3.7380	1.4260	0.0000	0.7330	0.6837	
T_DN23241	6	3.2690	1.3400	0.0000	0.6940	0.6411	
T_DN29012	6	2.9930	1.3660	0.0000	0.6660	0.6296	
T_DN20702	4	2.8250	1.1650	0.0000	0.6460	0.5805	
T_DN31208	3	2.0640	0.7640	0.0160	0.5160	0.3981	
T_DN24534	8	3.4960	1.5170	0.0000	0.7140	0.6796	
T_DN7799	3	2.0080	0.7200	0.0000	0.5020	0.3813	
T_DN20827	3	1.9270	0.8360	0.0000	0.4810	0.4300	
T_DN23877	7	2.8200	1.2460	0.0160	0.6450	0.5835	
T_DN27103	2	1.4200	0.4720	0.0050	0.2960	0.2522	
T_DN19607	5	1.4780	0.6560	0.0050	0.3230	0.2995	
T_DN26771	4	1.5400	0.7030	0.0050	0.3510	0.3285	
T_DN27978	6	2.4600	1.0530	0.0050	0.5930	0.5133	
T_DN25216	4	2.3480	0.9850	0.0090	0.5740	0.4872	
T_DN30193	3	1.4540	0.5950	0.0040	0.3120	0.2892	
T_DN9246	7	2.6130	1.3060	0.0000	0.6170	0.5846	
T_DN19478	2	1.3570	0.4330	0.0000	0.2630	0.2286	
T_DN34083	8	3.1450	1.3540	0.0000	0.6820	0.6263	
T_DN9805	8	2.2330	1.2340	0.0000	0.5520	0.5284	
T_DN30600	10	3.3960	1.4790	0.0000	0.7060	0.6562	
T_DN9218	5	3.4610	1.3330	0.0000	0.7110	0.6600	
T_DN23793	5	1.4620	0.6570	0.0000	0.3160	0.2960	
T_DN23793	4	2.9730	1.1610	0.0000	0.6640	0.5952	
T_DN30269	6	4.3940	1.5560	0.0000	0.7720	0.7348	
T_DN26655	4	2.8920	1.1930	0.0000	0.6540	0.5972	
T_DN18457	9	3.0000	1.4940	0.0000	0.6670	0.6402	
T_DN26075	9	4.4610	1.7740	0.0000	0.7760	0.7469	
T_DN27071	5	1.6390	0.8330	0.0000	0.3900	0.3713	
T_DN29286	4	1.8450	0.7390	0.0180	0.4580	0.3753	
T_DN32727	3	2.1770	0.9040	0.0000	0.5410	0.4700	
T_DN29213	3	2.3690	0.9370	0.0180	0.5780	0.4876	
T_DN33153	9	3.0670	1.4650	0.0000	0.6740	0.6327	
T_DN32381	4	1.5460	0.6940	0.0180	0.3530	0.3261	
T_DN33688	3	2.1570	0.8990	0.0310	0.5360	0.4670	
T_DN24007	4	2.1610	0.9210	0.0000	0.5370	0.4550	
T_DN25422	3	1.8120	0.7950	0.0000	0.4480	0.4057	
T_DN34120	3	2.5300	1.0120	0.0230	0.6050	0.5370	
T_DN29810	4	2.6610	1.0670	0.0410	0.6240	0.5471	
T_DN30324	6	4.2240	1.5610	0.0140	0.7630	0.7251	
T_DN26787	5	3.6250	1.3710	0.0310	0.7240	0.6748	
T_DN32610	4	2.3500	0.9870	0.0180	0.5740	0.4892	
T_DN16768	4	3.4530	1.3040	0.0050	0.7100	0.6572	
T_DN26565	4	1.8160	0.8400	0.0040	0.4490	0.4064	
T_DN31063	2	1.3790	0.4470	0.0090	0.2750	0.2370	
T_DN22109	4	1.6680	0.7480	0.0220	0.4010	0.3622	
T_DN28291	3	1.7430	0.6520	0.0130	0.4260	0.3427	
T_DN23988	4	1.8650	0.8650	0.0180	0.4640	0.4206	
T_DN24104	4	1.6920	0.7000	0.0270	0.4090	0.3501	
T_DN29493	7	3.7770	1.4850	0.0230	0.7350	0.6944	
T_DN25939	5	2.4380	1.0230	0.0140	0.5900	0.5070	
T_DN34009	6	3.6080	1.4600	0.0000	0.7230	0.6770	
T_DN25127	6	2.0840	1.0380	0.0360	0.5200	0.4772	
T_DN33367	5	3.6460	1.3650	0.0090	0.7260	0.6744	
T_DN28175	4	1.9680	0.8630	0.0200	0.4920	0.4311	
T_DN26387	3	1.2230	0.3710	0.0400	0.1820	0.1708	
T_DN25712	7	3.7830	1.5240	0.0000	0.7360	0.6950	
T_DN29236	5	1.9240	0.8470	0.0440	0.4800	0.4175	
T_DN30971	3	1.7930	0.6550	0.0090	0.4420	0.3484	
T_DN28375	3	2.2510	0.9240	0.1950	0.5560	0.4817	
T_DN23472	4	1.7540	0.8150	0.0000	0.4300	0.3904	
T_DN28326	9	5.3720	1.9020	0.0000	0.8140	0.7913	
T_DN24259	4	2.1160	0.8940	0.0230	0.5270	0.4550	
T_DN30467	5	2.8340	1.1990	0.0330	0.6470	0.5816	
T_DN30467	5	3.6560	1.3950	0.0000	0.7260	0.6789	
T_DN12372	5	1.6160	0.8020	0.0180	0.3810	0.3598	
T_DN25819	6	3.1940	1.4020	0.0000	0.6870	0.6474	
T_DN28542	6	2.2060	1.1320	0.0230	0.5470	0.5122	
T_DN17703	4	2.1390	0.9020	0.0090	0.5330	0.4591	
T_DN24002	4	1.9820	0.7580	0.0310	0.4950	0.3894	
T_DN30331	3	2.1380	0.8230	0.0090	0.5320	0.4247	
T_DN23145	4	1.8950	0.8570	0.0090	0.4720	0.4274	
T_DN33887	7	2.7570	1.3250	0.0000	0.6370	0.5997	
T_DN25920	6	2.1230	1.0190	0.0280	0.5290	0.4840	
T_DN30840	6	2.3740	1.0450	0.0000	0.5790	0.5144	
T_DN24924	2	1.6730	0.5920	0.0000	0.4020	0.3214	
T_DN20915	4	2.9090	1.1780	0.0370	0.6560	0.5903	
T_DN29632	5	2.0360	0.9590	0.0320	0.5090	0.4536	
T_DN15279	2	1.7850	0.6320	0.0050	0.4400	0.3430	
T_DN31937	3	2.0130	0.7250	0.0180	0.5030	0.3832	
T_DN22251	5	2.6600	1.1440	0.0000	0.6240	0.5630	
T_DN32482	6	2.6640	1.2160	0.2110	0.6250	0.5799	
T_DN32707	5	3.6570	1.3900	0.0050	0.7270	0.6782	
T_DN34099	2	1.1580	0.2630	0.0000	0.1370	0.1273	
T_DN29047	4	2.3340	0.9560	0.0000	0.5710	0.4801	

Based on the genotype data obtained from 103 pairs of SSR markers, the population genetic structure of 226 domestic faba bean materials was analyzed using Structure2.3.4 software. It has been shown that when K = 2, the value of ΔK is the maximum, and 226 samples were divided into two subgroups. A total of 194 (85.84%) samples, whose Q value >0.6, were divided into two groups. Group 1 contained 96 samples (42.48%) and Group 2 contained 98 samples (43.36%), respectively (Fig. 7). The Q value of the remaining 32 materials was ≤ 0.6 (14.16%). The result of Q value analysis revealed that there were no clear group attributive characteristics and a mixed population was formed.

Figure 7 The population structure of faba bean materials.

The yellow and blue colors indicate different groups.

Clustering analysis

A 0/1 matrix was established based on 498 loci amplified from 103 pairs of SSR primers with NTSYSpc Version 2.1 software. UPGMA clustering analysis of 226 faba bean germplasms was obtained (Fig. 8). The clustering results showed that 226 faba bean samples could be divided into five categories. Among these, Tongcan no.5 (H0005091; Jiangsu, China) and the beiyuanga faba bean (H0005277; Gansu, China) belonged to separate categories, respectively. Only two faba bean germplasms, including the erqing faba bean (H0001645) from the Sichuan province and big faba bean (H0001724) derived from the Guizhou province, were classified into the third category. The fourth class included 113 samples, which came from the majority of the faba bean planting provinces of China, including Sichuan, Gansu, and Qinghai. The remaining 109 materials were clustered into the fifth class.

Figure 8 UPGMA clustering of 226 faba bean varieties.

Results of population genetic structure analyses of 226 faba bean accessions with developed SSR markers.

Discussion

Our transcriptome analysis revealed 92.43 Gb of the overall sequencing data and provided a set of 133,487 faba bean unigenes using Illumina sequencing technology. A high-quality transcriptome assembly and annotation for faba bean was obtained in this study, which will provide a powerful tool for further gene mining of faba beans. Our analysis also showed the possibility of using the RNA-Seq technique for SSR development in faba bean. A total of 16,202 SSR loci and 5,200 developed SSR markers in this study will provide valuable information for genetic diversity, linkage mapping of genes with important agronomy traits, and marker-assisted selection in faba beans.

Faba beans are an important legume for human consumption and livestock feed, therefore, the planting area and production of faba bean have increased year by year. The planting range and yield of faba bean has ranked the fourth after pea (Pisum sativum L.), chickpea (Cicer arietinum L.), and lentil (Lens culinaris Medik.) according to FAO statistics (FAO, 2018).

Recently, more attention has been paid to the development of genomic and genetic resources of faba bean, owing to its increasing demands. The large genome size and its repetitive DNA (>85%) (Novák et al., 2020) have hampered the development of the faba bean in research areas including the sequencing of the whole genome, fine mapping, and map-based cloning.

Previous studies have reported transcriptome analysis and provided genetic resources for faba bean (Arun-Chinnappa & McCurdy, 2015; Ray, Bock & Georges, 2015; Braich et al., 2017; Cooper et al., 2017; Khan et al., 2019). Considerable progress has been made in some areas and previous studies have led to the DNA sequence data being released in public databases from previous reports (Mokhtar et al., 2020).

A genome-wide transcriptome map reported by Arun-Chinnappa & McCurdy (2015) generated 69.5 M reads and 65.8 M were used for assembly following trimming and quality control. A total of 17,160 unigenes were generated, of which 80.6% were successfully annotated against GO terms. Ray, Bock & Georges (2015) sequenced approximately 1.2 million expressed transcripts and assembled these into contigs by transcriptome analysis. Braich et al. (2017) used two Australian faba bean cultivars to characterize the transcriptome with RNA-Seq. The de novo assembly resulted in a total of 58,962 and 53,275 transcripts with approximately 67 Mbp (N50 Q 1,588 bp) and 61 Mbp (N50 Q 1629 bp) for two faba bean cultivars, respectively. Cooper et al. (2017) reported 16,300 unigenes by RNA-Seq to enhance the genome resources of faba bean. A total of 624.8 M high-quality reads were assembled into 198,155 unigenes with a mean length of 738 bp (N50 Q 1,347 bp) (Alghamdi et al., 2018).

The de novo assembly of 606.35 M high-quality pair-end clean reads yielded 164,679 unigenes of leaf tissues (Khan et al., 2019). Gene annotation regulated energy metabolism, transmembrane transporter activity, and secondary metabolites according to the GO and KEGG databases.

In this study, 133,487 unigene sequences with a total length of 178,152,541 bp, were assembled with N50 value of 1,858 bp. These results are higher in total number and average transcript length than previous studies (Arun-Chinnappa & McCurdy, 2015; Ray, Bock & Georges, 2015; Braich et al., 2017; Cooper et al., 2017; Alghamdi et al., 2018; Kareem et al., 2021). The number of unigenes (133,440) obtained in this study was also higher than other legume crops, such as lentil (20,419) (Kaur et al., 2011) and cowpea (47,899) (Chen et al., 2017a; Chen et al., 2017b). In addition, more unigenes (133,487) were annotated in seven databases (Fig. 2A) than previous studies (Kaur et al., 2012a; Kaur et al., 2012b; Braich et al., 2017; Khan et al., 2019). The groups that were most affected under the molecular function category of transcriptome data in this study were catalytic activity (39,008; 46.78%) and binding (35,451; 42.51%), which is similar to the results reported by Kareem et al. (2021).

The development of different sequencing technologies has led to an increasing number of databases and expressed sequence tags (ESTs), EST-SSRs, mtSSRs, microRNA-target markers, and single nucleotide polymorphism (SNP) markers reported in faba bean (Mokhtar et al., 2020). Thus, close genetic mapping, quantitative trait loci (QTL) mapping, and identification of candidate genes in faba bean are possible. EST-SSRs are a useful tool for faba bean genetic analysis compared with traditional SSRs because they are part of the genes. Additionally, EST-SSRs represent high conservation of expressed sequences and can transfer between related species.

In this study, we obtained 16,202 SSR loci from transcriptome analysis and developed a total of 5,200 EST-SSR markers from the 133,487 unigenes. These EST-SSR markers provide not only important genomic resources for basic research but also chances to construct closely linked mapping of agronomic traits for faba bean. The obtained SSR loci number in this study was lower than 18,327 loci reported by Alghamdi et al. (2018). However, the numbers of developed SSR markers was much more than previous reports, such as 2,932 SSRs revealed by Gnanasambandam et al. (2012) and 802 SSRs obtained by Kaur et al. (2012a); Kaur et al. (2012b). The SSR frequency and distribution analysis revealed that single base repeats were the dominant type of SSR sequences, accounting for 60.1% of the total number of SSR sequences. A/T had the largest number of repeats (9,261), accounting for 98.83% of single nucleotide repeats. The (AG/CT) n di-nucleotide repeat occurred 74.3%, which was the highest frequency in our study, and was similar to results reported by Kareem et al. (2021) and El-Rodeny et al. (2014). Simultaneously, SSR frequency and distribution in this study were consistent with the high (AG/CT) n repeats in other beans, such as the adzuki bean (Chankaew et al., 2014) and common bean (Blair et al., 2014). The results in this study also verified that SSR loci existed differently in various species and the screening criteria may also influence the distribution frequency of SSR loci.

To evaluate the polymorphism levels of the developed 5,200 EST-SSR markers, we selected 200 genic SSR loci with different nucleotide repeat types. One hundred and three primer pairs (51.5%) produced remarkable and repeatable bands between different faba bean varieties. The polymorphism ratio is lower than common bean (71.3%) (Hanai et al., 2007) and black gram (58.2%) (Souframanien, Reddy & Prasad, 2015), but higher than mung bean (33%) (Chen et al., 2015) and chickpea (47.3%) (Nayak et al., 2010).

In our study, the population genetic structure analyzed by Structure2.3.4 software showed that 226 faba bean samples were divided into two groups. However, the distance-based method clustered the samples into five groups. Our results revealed that the categories of germplasm from different geographical sources was not specially obvious. Faba bean materials from the same region were not able to be divided into the same group, and materials from different sources were clustered together, which was consistent with the results of Abid et al. (2015). This may be a result of gene exchange among germplasm resources from different regions and the mutual introduction of different regions in China. These results may also relate to the screening of SSR primers.

Conclusion

In conclusion, this transcriptomic analysis of four faba bean varieties with different phenotypes provided a valuable set of genomic data for characterizing genes. In addition, the developed SSR markers in this study will lay a basis for further genetic studies and molecular marker-assisted breeding of faba bean.

Supplemental Information

Supplemental Information 1 226 faba bean varieties derived from different provinces of China

Unified national number, variety name and source name of 226 faba bean varieties.

Click here for additional data file.

Supplemental Information 2 Raw data calculated with the RNA Seq-Power package in R

The sequencing depth was 32 and the sample number was 3

Click here for additional data file.

Supplemental Information 3 Overview of the RNA-seq data

Click here for additional data file.

Supplemental Information 4 Quality index of unigenes

Analysis of unigenes

Click here for additional data file.

Supplemental Information 5 Sequences of all 5200 developed SSR markers

5200 SSR markers. SSR motif and repeats were analyzed with Microsatellite identification tool and the primers were designed using Primer 3.0.

Click here for additional data file.

Supplemental Information 6 Sequences of 103 polymorphic SSR markers used for genetic diversity analysis

Sequences of developed SSR markers. SSR motif and repeats were analyzed with Microsatellite identification tool and the primers were designed using Primer 3.0.

Click here for additional data file.

We are grateful to Dr. Yang at CAAS in Beijing, China for providing the germplasm resources for our study.

Additional Information and Declarations

Competing Interests

Author Contributions

Data Availability

The authors declare there are no competing interests.

Wanwei Hou conceived and designed the experiments, performed the experiments, analyzed the data, prepared figures and/or tables, and approved the final draft.

Xiaojuan Zhang performed the experiments, prepared figures and/or tables, and approved the final draft.

Yuling Liu analyzed the data, prepared figures and/or tables, and approved the final draft.

Yujiao Liu analyzed the data, authored or reviewed drafts of the article, and approved the final draft.

Bai li Feng conceived and designed the experiments, authored or reviewed drafts of the article, and approved the final draft.

The following information was supplied regarding data availability:

The raw sequencing data are available at the National Center for Biotechnology Information (NCBI): PRJNA823658; SAMN27335281; SAMN27335282; SAMN27335283; SAMN27335284; SAMN27335285; SAMN27335286; SAMN27335287; SAMN27335288; SAMN27335289; SAMN27335290; SAMN27335291; SAMN27335292.

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
