# Peer review of "RNA-Seq and genetic diversity analysis of faba bean (Vicia faba L.) varieties in China"

_PeerJ, doi:10.7717/peerj.14259_

## Round 0.1 · original submission · Major Revisions

Comments are attached below:

·

Basic reporting

In this paper “Genetic diversity analysis of faba bea (Vicia faba L.) varieties in China based on developed SSR markers with RNA-Seq technology” the authors investigated the genetic diversity of 226 indigenous faba bean varieties through SS markers. This is an interesting work and the data generated is valuable for scientific community. However, the manuscript needs extensive revisions before considering it for publication
The manuscript has language and editing issues and revision is required. In some lines, the inappropriate use of English language makes the manuscript really hard to follow. There are repeated and unnecessary statements at some places. Some comments are as follows:
- Title should be more precise and understandable. Spelling of faba bean is incorrectly written as faba bea in the title.
- Abstract needs substantial improvement; there is no proper background and methodology. There are lots of English language issues that makes the abstract really hard to follow. For example, there is repetition of themes/statements. On line 26-27, the statement “RNA-Seq will provide a feasible approach to develop SSR markers for both genetic diversity and mapping analysis for further faba bean” is meaningless and incomplete. Result part should be more elaborative. Authors are suggested to add all core findings of their study in the abstract. Authors should use full name instead of abbreviation in keywords section.
- The Introduction is well-written and has sufficient background information.
- Figures are relevant and have good resolution. Overall, the text in figures is readable except figure 6. Figure 2A and 2B are inversely cited in the text. Authors are advised to recheck all figure citations carefully throughout the manuscript.
- Authors have provided raw data as per journal requirement.
- In text citation and reference section also need revision. For example, on third paragraph of discussion, in text citation of Kareem A. Khalifa et al(2021) is incorrectly cited. Similarly, scientific name of “Vicia faba” in 3rd reference in the reference section must be in italic.

Experimental design

This is an original research work and fall well within the scope of the journal. Experimental design is valid and justifies the work need. Overall, the methodology section is well written and meets the ethical and technical standards.

Validity of the findings

The result part is well written and validating the information generated during this research. Figures description is good and is easy to follow. However, there are some spacing issues in “SSRs frequency and distribution” section. As for as the discussion section is concerned, the first two paragraphs of discussion are written more like results. Overall, the discussion section is poorly written and the authors are suggested to make extensive revision of this section and they need to incorporate more recent and relevant references in this section e.g. Yang et al., 2017. Conclusion part must be rewritten. In its current form, it is just repetition of results.

Additional comments

Overall, the manuscript is technically sound and has additional findings in the existing pool of knowledge. However, based on the queries raised in above parts, the manuscript needs extensive revisions before considering it for publication.

Reviewer 2 ·

Basic reporting

The article by Hou et al. presents an RNA-seq analysis, performed on 4 faba bean genotypes, for the identification of polymorphic SSRs, and subsequent genotyping of these markers in a collection of 226 individuals of the species. The rationale for using the selected methodology for marker development is that the reference genome for the species is not yet available, given its large size (~13 gb).

I attach my comments on this work, separated by sections (introduction, M&M, results). I consider that the work requires substantial changes to be published.

Introduction: I recommend an updated review of the available literature and material for the species. The most recent article cited in the introduction dates from 2018. Since then to date, great advances have been made in the development of genomic tools for the species, some of which could be used to compare with the results obtained here (e.g. https://doi.org/10.1038/s41598-021-00506-0, who did long read RNAseq, and https://doi.org/10.1093/aobpla/plaa064, who present the development of an EST, EST-SSR and mtSSR database for the species, among many other articles). Thus, I suggest rewriting the introduction section bounded by lines 61-102, updating with recent literature.

Experimental design

M&M: in the section 'Data filtering and de novo assembly', please provide for each software the parameters used.

M&M: I really don't understand the paragraph on lines 153-155. What does it mean that the sample number was 3. Wasn't it 4 accessions?

M&M: I also don't understand the following paragraph (lines 156-160): 'The sequencing data were available at the National Center for Biotechnology Information (NCBI) database'. What data was already available at NCBI? Didn't you generate them in this work? Or did you also used public data?

M&M: in the 'Genetic diversity analysis' section, STRUCTURE parameters (burn-in, etc, which is presented in results but not in M&M) are missing.

Validity of the findings

Results: in the 'RNA sequencing and Sequence annotation' section, what was the length of the sequencing reads? Were these paired-end or single-end? What percentage of these were assembled to generate genes? How was the filtering of 'redundant' genes done? Add confidence intervals (or ranges) for the sizes of the assembled unigenes.

Results: In the 'Gene function classification' section. You could present the numbers of genes annotated by each database in a table, along with the main categories annotated. The text is lost, confusing, and the bar chart (Figure 2A) doesn't help. A table is more informative and sorts the information better. Also, how many of the 133,487 unigenes were annotated by at least one database?

Results: In the section 'Gene function classification' (lines 216-226). Please elaborate a little more on how the GO/ Blast2Go results are interpreted. The interpretation given is not correct.

Results: I do not understand why the authors do the annotation with the 7 databases, if in the following section ('SSRs frequency and distribution ') they continue working with the 133,487 previously assembled unigenes. I understand that the annotation, besides helping to describe the transcriptome of the species (which is already done, by previous authors), provides the authors with robustness and evidence of how well assembled the transcripts are. Thus, I recommend doing this analysis and subsequent steps only using the unigenes that were annotated by at least one database, i.e. those for which there is more robust evidence.

Results: In the section 'SSRs frequency and distribution', why do the authors make a random selection of the 200 SSR markers? Shouldn't it be fairly targeted, to obtain fragments that maximise the number of alleles among the 4 sequenced genotypes, with more elegant repetitive motifs (discarding mononucleotides), and that generate amplification bands of different sizes, to facilitate genotyping (more if done in gel) and/or future multiplexing? Or that they fall on genes of interest (taking advantage of the fact that they are annotated).

Results: in the section 'SSR polymorphism analysis'. The authors mention on the Structure output that k=2 contains higher value of deltaK, but that the distribution of genotypes in two subpopulations does not correlate with known underlying structure. They should evaluate the next most likely k value. It is known that the peak at k=2 in Structure may be artifactual (https://doi.org/10.3732/ajb.0800097). And see if this new value of K correlates with any underlying structure.

Results: in the 'SSR polymorphism analysis' section. Report on filtering of the marker matrix. Maximum percentage of missing data accepted, filtering by MAF, etc. If these filterings were not done, they should be done and returned to the 'SSR polymorphism analysis' section.

Results: in the section 'SSR polymorphism analysis'. Authors should report on the filtering applied on the marker matrix (i.e. maximum percentage of missing data accepted, filtering by MAF, etc). If such filtering was not done, it should be done and the data re-analysed (as alleles with very low frequency may be genotyping errors and not true alleles, which may affect and/or bias population analyses).

Results: Add bootstrap values in Figure 9. In addition, the authors should compare the classification results obtained by this distance-based method vs. the clustering obtained by STRUCTURE.

Results: I consider that the article is too long. Some figures can go as supplementary material.

Discussion: Accommodate the discussion after incorporating the suggested changes.

Additional comments

IMPORTANT I: I recommend doing a thorough review of the writing with an English speaker. There are many sentences that are not properly understood and/or appear 'cut off'.

IMPORTANT II: The data (both raw sequencing data and fasta data containing the assembled unigenes) should be deposited in public databases.

Reviewer 3 ·

Basic reporting

This study employed faba bean transcriptome information to develop SSR markers to assess a Chinese faba bean germplasm set. In previous studies [Duc et al., 2010; Field Crops Research 115 (2010) 270–278] the Chinese faba bean germplasm revealed a wide genetic diversity and due to unknown faba bean wild species measuring and capturing this diversity is important.
I have a fundamental question, why authors developed gel-based SSR markers and not SNP! Around myself, at least, nobody is a fan of SSR markers anymore!

Experimental design

I have a fundamental question, why authors developed gel-based SSR markers and not SNP! Around myself, at least, nobody is a fan of SSR markers anymore!
Literature review (introduction) is widely missing many updated published works in faba bean genomics (see https://doi.org/10.1002/leg3.75). The introduction needs to be updated with recent references.

Validity of the findings

no comment

Additional comments

L. 57, " 13.4 Gb" is not correct, should be around 13.
L. 118, Please provide more details for those lines, I mean line-specific, which one was, for example, low tannin, etc.
L. 123, "materials" what kind of material?
- The authors should explain how and why these 226 accessions were selected for this study.
200 (103 polymorphic) out of 5200 SSRs were used to assess the genetic diversity? It was mentioned this was selected randomly?? I think you could select the markers spanning the whole six faba bean chromosomes?
I would suggest authors provide the sequence of all 5200 markers to readers.

---

## Round 0.2 · accepted · Accept

Authors are requested to incorporate minor format changes, as suggested by a reviewer.

·

Basic reporting

I have gone through the revised version of manuscript entitled "RNA-Seq and genetic diversity analysis of faba bean (Vicia faba L.) varieties in China". The authors have made satisfactory changes in the revised version of the manuscript as per suggestions. The manuscript in its current form can now be ACCEPTED for publication in your prestigious journal.

Experimental design

no comment

Validity of the findings

The authors have made substantial changes in the results and discussion section and these parts are much improved now. Conclusion is also amended. The authors have generated some interesting results in this study and these will be helpful for future research.

Reviewer 4 ·

Basic reporting

The authors of this study provide an RNA-seq analysis of 226 faba bean (Vicia faba L.) varieties for the identification of polymorphic SSRs and subsequent genotyping of these markers. Due to the unknown faba bean wild species, Chinese faba bean germplasm revealed a significant genetic variation, making it crucial to measure and capture this diversity. The data produced as a result of this intriguing endeavor is useful to the scientific community. Before evaluating the work for journal publication, however, some small adjustments are required. Please seek the assistance of a language specialist for manuscript editing and proofreading as this article has some linguistic difficulties.

Experimental design

This is an original research work and falls well within the scope of the journal. Experimental design is valid and justifies the work needed. Overall, the methodology section is well written and meets ethical and technical standards.

Validity of the findings

This research work has generated validated results. Figures are nice and ligands are also elaborative. Please try to improve the discussion portion and compare your findings with some standard and validated datasets as well.

Additional comments

1- In the reference section please format the bibliography as per the journal's style, some references have spacing issues i.e. Alghamdi SS, Migdadi HM, Ammar MH, Paull JG, Siddique KHM. 2012. Faba bean genomics: Current status and future prospects. Euphytica 186:609–624,
DOI 10.1007/s10681-012-0658-4 and Altschul SF, Madden TL, Schäffer AA, Zhang J, Zhang Z, Miller W, Lipman DJ. 1997. Gapped BLAST and PSI-BLAST: a new generation of protein database search programs, Nucleic Acids Res .25(17)3389-3402,
DOI 10.1093/nar/25.17.3389. There are a number of such formatting errors.
2- Authors have chosen 'Times new Romans font style' to annotate figure 1, but in figure 2 they have selected 'Calibri font style'. Try to be consistent while annotating figures and enlarge the color key Venn diagram in the figure. In figure 5, names of genes are not properly written italicize them and make them clear. Enhance the resolution of figure 8 'phylogenetic tree'.
3- As public databases are synchronized consistently for data accuracy check, mention data accessed dates with all public databases data retrieval in M.M.